# Resveratrol Treatment Enhances the Cellular Response to Leptin by Increasing OBRb Content in Palmitate-Induced Steatotic HepG2 Cells

**DOI:** 10.3390/ijms20246282

**Published:** 2019-12-12

**Authors:** Andrea Ardid-Ruiz, Maria Ibars, Pedro Mena, Daniele Del Rio, Begoña Muguerza, Lluís Arola, Gerard Aragonès, Manuel Suárez

**Affiliations:** 1Nutrigenomics Research Group, Department of Biochemistry and Biotechnology, Universitat Rovira i Virgili, 43007 Tarragona, Spain; 2Human Nutrition Unit, Department of Food and Drugs, University of Parma, 43125 Parma, Italy; 3Human Nutrition Unit, Department of Veterinary Medicine, University of Parma, 43125 Parma, Italy; 4School of Advanced Studies on Food and Nutrition, University of Parma, 43215 Parma, Italy; 5Microbiome Research Hub, University of Parma, 43125 Parma, Italy; 6Eurecat, Centre Tecnològic de Catalunya, Biotechnological Area, 43204 Reus, Spain

**Keywords:** leptin resistance, lipid metabolism, non-alcoholic fatty liver disease (NAFLD), obesity, resveratrol metabolites, sirtuin 1

## Abstract

The interaction of leptin with its hepatic longest receptor (OBRb) promotes the phosphorylation of signal transducer and activator of transcription-3 (STAT3), protecting the liver from lipid accumulation. However, leptin signalling is disrupted in hepatic steatosis, causing leptin resistance. One promising strategy to combat this problem is the use of bioactive compounds such as polyphenols. Since resveratrol (RSV) is a modulator of lipid homeostasis in the liver, we investigated whether treatment with different doses of RSV restores appropriate leptin action and fat accumulation in palmitate-induced steatotic human hepatoma (HepG2) cells. Both RSV metabolism and the expression of molecules implicated in leptin signalling were analysed. RSV at a 10 μM concentration was entirely metabolized to resveratrol-3-sulfate after 24 and counteracted leptin resistance by increasing the protein levels of OBRb. In addition, RSV downregulated the expression of lipogenic genes including *fatty acid synthase* (*Fas*) and *stearoyl-CoA desaturase-1* (*Scd1*) without any significant change in Sirtuin-1 (SIRT1) enzymatic activity. These results demonstrate that RSV restored leptin sensitivity in a cellular model of hepatic steatosis in a SIRT1-independent manner.

## 1. Introduction

Dietary bioactive compounds are currently being investigated to complement therapeutic strategies to prevent or combat many diseases including non-alcoholic fatty liver disease (NAFLD) and leptin resistance [1,2,3]. In this sense, functional foods are gaining attention and more research is continually being performed to support new health claims. Within this context, we have previously reported that a polyphenol-rich extract from grape seeds protects against hepatic fat accumulation by increasing the availability of cellular nicotinamide adenine dinucleotide (NAD^+^) and the functionality of sirtuin 1 (SIRT1) [4]. In addition, resveratrol (3,5,4’-trihydroxystilbene, RSV), a dietary non-flavonoid polyphenol found in grapes (0.002–0.008 mg/g) and red wine (1.98–7.13 mg/L) has also been found to protect the liver against lipid metabolic disorders in various rodent models of hepatic steatosis, possibly via regulating SIRT1 activity and endoplasmic reticulum (ER) stress [5,6,7]. In fact, RSV has been indicated to have a wide range of biological effects, including cardioprotective, anticancer, anti-inflammatory, and antioxidant properties [8,9]. However, the exact mechanisms by which RSV exerts its beneficial effects are still unclear and require more research.

Several studies have shown that exogenous leptin administration regulates hepatic lipid and glucose homeostasis in rodent models of dysfunctional leptin signalling in *ob*/*ob* and *db*/*db* mice [10,11]. This suggests that hepatic leptin action is essential for proper fat and carbohydrate metabolism in this tissue. At the molecular level, when leptin interacts with its hepatic longest receptor isoform (OBRb), it promotes the phosphorylation of signal transducer and activator of transcription-3 (STAT3) through the activation of Janus tyrosine kinase 2 (JAK2). Subsequently, STAT3 dimerizes and translocates into the nucleus [12], downregulating hepatic lipogenesis and upregulating fatty acid oxidation [13]. Hence, defects in hepatic leptin action, which occur in the state of diet-induced obesity, impair the function of the liver, leading to fat accumulation [8,11]. The molecular basis for this lack of leptin response in the liver is not yet completely known, but it has been attributed to several mechanisms. These include OBRb insensitivity, enhanced ER stress and inflammation, impaired NAD^+^-dependent deacetylase SIRT1 activity and the overexpression of inhibitory factors such as suppressor of cytokine signalling 3 (SOCS3) and protein tyrosine phosphatase (PTP1B) [11,12,14,15,16].

Considering the promising results of RSV against lipid metabolic disorders and given that little is known about how RSV specifically affects the regulation of leptin signalling in the liver, the aim of the present study was to evaluate the effects of this compound on the cellular response to leptin in palmitate-induced steatotic human hepatoma (HepG2) cells. Additionally, the capacity of HepG2 cells to metabolize RSV was also explored in order to gain insights into the fate of RSV in both non-steatotic and palmitate-induced steatotic HepG2 cells [17].

## 2. Results

### 2.1. RSV Improves Leptin Signalling and Lipid Content in Steatotic HepG2 Cells

To evaluate whether RSV could sensitize steatotic HepG2 cells to leptin, cells were treated with or without RSV (1, 5, 10, 50 μM) for different time intervals (20 min, 6 h, and 24 h) and were exposed to 10 ng/mL leptin during the last 20 min to activate the leptin pathway. Notably, the treatment of HepG2 cells with RSV for 20 min had no effect on pSTAT3 levels in any dose (Figure 1A) and even 10 μM of RSV significantly increased triglyceride (TAG) content compared to untreated cells (Figure 1B). After 6 h of treatment, only the dose of 50 μM RSV was able to increase STAT3 phosphorylation (Figure 1C), although no significant differences were observed in TAG accumulation (Figure 1D). By contrast, after 24 h of treatment, both 10 and 50 μM RSV significantly restored pSTAT3 levels to values similar to those observed in non-steatotic cells (Figure 1E), indicating that RSV could rescue leptin sensitivity in our cellular model. In addition, after 24 h of treatment with RSV, the intracellular concentrations of TAG decreased significantly to basal levels with all the doses tested, reversing the lipid accumulation observed in the untreated cells, although the effects of treatment with the 50 μM dose did not reach statistical significance (Figure 1F). It should be noted that the exposure of steatotic HepG2 cells to the highest concentration of RSV for 24 h did not decrease cell viability compared to that of untreated cells (Appendix A).

### 2.2. RSV Is Rapidly Metabolized into RSV-3-Sulfate

To evaluate the metabolism of RSV in steatotic HepG2 cells and to gain insights into the molecules which appeared after 24 h of treatment, the cell media was analysed by ultra-high-performance liquid chromatography–mass spectrometry (UHPLC-MSn) at the beginning and end of each incubation (20 min, 6 h, and 24 h) of HepG2 cells. In addition, non-incubated cell media were used to test RSV stability in the absence of cells. Different metabolic reactions, including conjugation with glucuronide and sulphate moieties, and combinations thereof, were monitored. Notably, RSV was entirely conjugated to RSV-3-sulfate (R3S) at a concentration of 10 μM after 24 h of incubation with cells (Figure 2A). The proportion of R3S to the total RSV in steatotic HepG2 cells was approximately 5% after 20 min of incubation, increasing to 45% after 6 h and reaching almost 100% after 24 h. The incubation of RSV in non-steatotic HepG2 cells did not differ with respect to the metabolic activity observed in our experimental model of steatosis since the ratio of RSV to R3S was almost equal to that observed in steatotic cells (Figure 2B).

### 2.3. RSV Modulates Lipogenic Gene Expression but Not Fatty Acid Oxidation

When the mRNA levels of key genes involved in lipid metabolism were assessed by RT-qPCR, treatment with 10 μM RSV for 24 h resulted in decreased *fatty acid synthase* (*Fas*) and *stearoyl-CoA desaturase-1* (*Scd1*) mRNA levels compared to those measured in the untreated cells. This may indicate that the delipidating effect of RSV is mediated by limiting the capacity for de novo lipogenesis (Figure 3A). In contrast and contrary to our expectations, no significant changes were observed in the expression of genes that encode enzymes for fatty acid oxidation (Figure 3B).

### 2.4. RSV Does Not Modulate the Expression of Pro-Inflammatory and ER Stress-Related Genes

In our attempt to determine the molecular mechanism by which RSV enhances the cellular response to leptin, we assessed the expression of genes involved in ER stress and inflammation by qRT-PCR (Figure 4A). Although *Activating transcription factor 4* (*ATF4*) expression was downregulated after 24 h of RSV treatment compared to its expression in control cells, no significant differences were found in *DNA damage inducible transcript 3* (*CHOP*) and *spliced x-box binding protein 1* (*SXBP1*), indicating that supplementation of the cell media with palmitate (Palm) + glucose (Glc) for 48 h did not induce ER stress in HepG2 cells, and, in turn, that these transcripts were not significantly affected by RSV after 24 h of treatment. In a similar manner, transcripts related to inflammation such as inducible *nitric oxide synthase* (*iNOS*), *interleukin-6* (*IL-6*), and *tumour necrosis factor alpha* (*TNF-α*) were not significantly modulated by RSV treatment.

In addition, to further investigate the effects of RSV treatment on the leptin signalling pathway, the mRNA expression of negative regulatory molecules involved in STAT3 activation such as *SOCS3* and *PTP1B* was also assessed. However, RSV did not revert the increased expression levels of *SOCS3* observed in steatotic cells and *Ptp1b* mRNA levels were not statistically altered under any experimental condition (Figure 4B).

### 2.5. RSV Increases the mRNA and Protein Levels of Leptin Receptor OBRb but Not SIRT1 Activity

Finally, we evaluated whether RSV treatment could enhance SIRT1 activity, which is an additional mechanism involved in the regulation of leptin signalling in hepatic cells [7]. However, the deacetylase activity of SIRT1 was not significantly affected after 24 h of treatment with RSV at any dose (Figure 5A). Hence, we determined by immunoblotting whether pSTAT3 was directly mediated by an increase in the cellular content of the long leptin receptor isoform OBRb. Interestingly, RSV treatment at all doses for 24 h increased the protein levels of OBRb, although statistically significant differences were only observed at 5 and 10 μM RSV (Figure 5B). These results were further confirmed by immunocytochemistry using a specific anti-OBRb antibody. Remarkably, OBRb expression was higher in steatotic HepG2 cells treated with 10 and 50 μM RSV than in untreated cells (Figure 5C).

## 3. Discussion

In the present study, we found that RSV increased the cellular response to leptin by increasing the OBRb content in steatotic HepG2 cells and reduced the intracellular lipid content without any significant change in SIRT1 activity, inflammatory status, or ER stress markers. Previous studies have already shown the relevance of leptin receptor/STAT3-dependent signals in the regulation of lipid homeostasis [18], as well as the effect of RSV on hyperleptinaemia by increasing pSTAT3 content in the hypothalamus of adult offspring from high-fat-obese rats [19] and in the metabolic tissues of cafeteria-obese rats [7]. However, to the best of our knowledge, this article provides the first instances of evidence regarding the molecular mechanisms by which these beneficial effects of RSV are achieved in hepatic cells. In addition, we also identify the most abundant metabolite generated after treatment with RSV, providing a better understanding of what is happening in the cell culture as a response to the RSV treatment.

To evaluate in vitro the impact of RSV on the leptin signalling pathway, it was first necessary to develop a suitable experimental model in HepG2 cells. For this purpose, and based on our previous work [20], we determined that the leptin pathway is most active when cells are incubated for 20 min with 10 ng/mL leptin (Appendix A) and that the combination of 0.5 mM Palm + 30 mM Glc for 48 h is the most effective treatment in causing a state of leptin resistance by reducing pSTAT3 levels and increasing TAG accumulation (Appendix A). Once we established conditions to induce leptin resistance in cultured cells, we evaluated the ability of RSV to restore the protein levels of pSTAT3 and the TAG content after different incubation times. Our results confirmed that RSV acts as a leptin sensitizer only in a long-term manner, because only after 24 h of incubation was RSV able to properly restore STAT3 activation and TAG accumulation to their basal levels. These results suggest that the delipidating effect of RSV is at least partially mediated by the increase in pSTAT3 and therefore that leptin signalling is crucial to maintaining correct lipid metabolism in these cells. Interestingly, this normalization was associated with the capacity of RSV to downregulate *Fas* and *Scd1* gene expression. In this sense, a recent study has demonstrated the ability of brain leptin to protect from hepatic steatosis by decreasing de novo lipogenesis in rat liver [21]. In fact, our results indirectly confirmed this finding, probably due to the effect of RSV on increasing the cellular response to leptin. In addition, the lack of leptin-induced changes in hepatic gene expression of peroxisome proliferator-activated receptor α (*Pparα*) and carnitine palmitoyltransferase 1a (*Cpt1a*), which are regarded as important regulators of lipid oxidation, did not suggest a major role for hepatic β-oxidation in our experimental treatment with RSV. Other studies carried out in HepG2 cells using higher amounts of glucose, palmitic acid, or oleic acid to induce fat accumulation have also reported the beneficial effects of both RSV and other (poly)phenols. These studies concluded that the antisteatotic effects of these molecules were a consequence of not only the downregulation of the expression of genes involved in the lipogenesis pathway but also the activation of 5’AMP-activated protein kinase (AMPK) [22,23,24,25]. These results reinforce the idea that specific polyphenols improve hepatic fat accumulation by different mechanisms.

The induction of leptin resistance has been attributed to inflammation as a result of the induction of pro-inflammatory signalling molecules such as JNK and NF-κB and ER stress following fat accumulation [12,14]. However, in the present study, despite this improvement of leptin signalling, cells treated with RSV did not display a significant decrease in markers of inflammation and ER stress, indicating that RSV at the dosage used in this experiment was not sufficient to completely reverse these cellular alterations. These data are consistent with previous results obtained in cafeteria Wistar rats supplemented with different doses of RSV (25, 100, and 200 mg/kg body weight) in which we did not find significant differences in hepatic inflammatory and ER stress markers [7]. Instead, in this previous study [7], SIRT1 activity was highlighted as a mediator of leptin action in this tissue.

SIRT1 is an important modulator of hepatic lipid metabolism by enhancing fatty acid oxidation, decreasing lipogenesis, and modulating cholesterol levels through the activation of AMPK [4]. However, no significant changes in SIRT1 activity and gene expression were observed in the present study, indicating that if RSV had the ability to enhance the cellular response to leptin in our experimental model, it was not mediated by an increase in SIRT1 functionality. Thus, the experimental model and the grade of hepatic damage achieved in this study could affect the impact of RSV in these cells, and, therefore, further in vitro experiments using other less-potent inductors or a shorter time of incubation with these inductors should be performed to confirm these results. In addition, it should be noted that we worked with a human hepatoma-derived cell line which means that the behaviour could be different if we compared behaviours with respect to non-cancerous cells.

OBRb is also a mediator of leptin action in both central and peripheral organs [12]. Thus, the enhanced OBRb protein content induced by RSV reported for the first time in the present study could be the mechanism by which this compound increased the cellular response to leptin in our experimental model of hepatic fat accumulation. In fact, we have demonstrated in previous studies that cafeteria-fed rats supplemented with a high dose of RSV show an increase in OBRb protein concentrations in skeletal muscle [7] and in rat brain endothelial cells [26], but it has never been reported in the liver. In the present study, the effects of RSV on the cellular content of OBRb assessed by Western blotting and immunofluorescence suggest that RSV re-established the leptin sensibility of steatotic hepatocytes by increasing the cell surface content of ObRb to enhance the cellular response to leptin. In this context it is also important to point out that although there could be considerable differences in OBRb content between cancerous and non-cancerous cells, with regard to regulation of the hepatic metabolism, HepG2 cells appear to be closer to the in vivo situation despite their tumorigenic origin [27]. However, since we only documented this for HepG2 cells, future studies are needed to evaluate whether this mechanism is also at work in non-cancerous hepatic cells.

Finally, the capacity of HepG2 cells to metabolize RSV was also explored in the present study. Interestingly, our data suggested that under these experimental conditions, these cells only have the ability to conjugate free RSV into RSV-3-sulfate in less than 24 h. Although RSV glucuronidation may occur in the human liver [28], our analysis did not show any evidence of glucuronide conjugation. These results are in accordance with previous studies reporting that RSV conjugation occurs rapidly and mostly as sulphate conjugates (twice more than glucuronides) in human cells [29,30]. In addition, RSV was stable in the absence of cells, which accounts for the lack of spontaneous degradation and the complete availability of RSV for the cells. This stability is also consistent with results obtained in human endothelial cells incubated with different forms of RSV [31].

## 4. Materials and Methods

### 4.1. Chemicals

Dulbecco’s modified Eagle medium (DMEM), penicillin/streptomycin (P/S), L-glutamine (*L*-Gln), and foetal bovine serum (FBS) were purchased from Lonza (Barcelona, Spain). Non-essential amino acids (NEAA), amphotericin B, 4-(2-hydroxyethyl)-1-piperazineethanesulfonic acid (HEPES) buffer, Palm, non-essential fatty acid (NEFA), free bovine serum albumin (BSA), 2-mercaptoethanol, and CaCl_2_ were obtained from Sigma Aldrich (Madrid, Spain). Glc, neutral red dye, and glacial acetic acid were purchased from Panreac AppliChem (Barcelona, Spain). RSV was purchased from Fagron Iberica (Barcelona, Spain) and resveratrol-3-sulfate was obtained from Bertin Pharma (Montigny le Bretonneux, France). Formaldehyde and ethanol were obtained from Millipore (Madrid, Spain). All LC-MS-grade solvents were purchased from Sigma-Aldrich (St. Louis, MO, USA). Finally, recombinant human leptin was obtained from BioVision (San Francisco, CA, USA).

### 4.2. Cell Culture and Experimental Design

HepG2 cells (HB-865; ATCC, Richmond, VA, USA) were cultured in DMEM containing 0.1 mM NEAA, 100 U/mL penicillin, 100 mg/mL streptomycin, 250 mg/L amphotericin B, 2 mM *L*-Gln, 10% (*v*/*v*) FBS, and 1.25 M HEPES at 37 °C in a humidified atmosphere containing 5% CO_2_ with medium changes three times a week. Cells were incubated in 12-well plates at a density of 5 × 10^5^ cells/well for 48 h or until they were 70–80% confluent before starting the experimental treatments. Hepatic steatosis was induced by simultaneously treating cells with 0.5 mM Palm and 30 mM Glc (Palm + Glc) for 48 h using BSA as a fatty acid carrier (7:1 Palm:BSA). After successfully producing the hepatic steatosis model, cells were treated with or without RSV (1, 5, 10, and 50 μM) for different time intervals (20 min, 6 h, and 24 h). Afterwards, cells were starved for 16 h with FBS-free medium and then exposed to 10 ng/mL recombinant human leptin for 20 min.

### 4.3. Palmitate-BSA Solution Preparation

The Palm-BSA complexes were prepared as previously described [32], with minor modifications. Briefly, 13.9 mg sodium Palm was dissolved in 0.5 mL sterile water (100 mM Palm stock solution) by heating (70 °C) and mixing (250 rpm) for 10 min in a thermomixer (Grant-Bio). A portion of the Palm stock solution (50 µL) was added to 950 µL BSA-free DMEM containing 5% NEFA-free BSA (the 5 mM Palm working solution). The Palm working solution was also heated (40 °C) and vigorously shaken (250 rpm) for 1 h. Finally, the working solution was filtered (20 nm diameter filter) and immediately used to treat the cells. BSA-free DMEM containing 5% NEFA-free BSA was used as the control vehicle.

### 4.4. Determination of Cell Viability by Neutral Red Assay

Cells were incubated with neutral red (NR) dye to assess toxicity as previously described [33,34]. Briefly, 1 mL of freshly prepared NR solution (0.05 mg/mL) was pre-warmed to 37 °C and added to each well (in a 12-well plate). The cells were then incubated for 3 h at 37 °C. Then, the dye was removed and the cells were exposed to 1 mL/well of fixative solution (CaCl_2_ and 37% formaldehyde). After washing, the cells were incubated with 1 mL of a decolorizing solution (1% glacial acetic acid and 50% absolute ethanol). Following 10 min of shaking at room temperature to release the dye from the cells, the absorbance was measured at 540 nm using an automatic plate reader (EON Microplate, BioTek, Winooski, VT, USA).

### 4.5. Quantification of Triglyceride Content

The triglyceride concentration in the cell lysates was measured using the GPO enzymatic colorimetric assay (QCA, Barcelona, Spain) according to the manufacturer’s instructions. All results were normalized to the total protein content using the BCA method and calculated with respect to the control group.

### 4.6. SIRT1 Activity Assay

SIRT1 activity was determined using a SIRT1 direct fluorescent screening assay kit (Cayman, Ann Arbor, MI, USA) as previously described [7]. Briefly, 25 μL of assay buffer (50 mM Tris-HCl, pH 8.0, containing 137 mM NaCl, 2.7 mM KCl, and 1 mM MgCl_2_), 5 μL of cell lysate, and 15 μL of substrate (Arg-His-Lys-Lys(ε-acetyl)-)-7-amino-4-methylcoumarin) was added to all wells. The fluorescence intensity was monitored every 2 min for 1 h using the fluorescence plate reader Berthold Tech TriStar2S (Berthold Technologies, Bad Wildbad, Germany) with an excitation wavelength of 355 nm and an emission wavelength of 460 nm. The results were expressed as the reaction rate for the first 30 min as there was a linear correlation between the fluorescence and time over this period.

### 4.7. Total RNA Isolation and Gene Expression Analysis

Total RNA was obtained from the cells using the RNeasy Mini Kit (Qiagen, Valencia, CA, USA). A NanoDrop 1000 spectrophotometer (Thermo Scientific, Madrid, Spain) was used to determine sample concentration and purity. A High-Capacity cDNA Reverse Transcription Kit (Thermo Fisher, Madrid, Spain) was used to generate the cDNA using 500 ng of RNA. Obtained cDNA was subjected to qPCR with an iTaq Universal SYBR Green Supermix (Bio-Rad, Barcelona, Spain) using the 7900HT Real-Time PCR system (Applied Biosystems, Foster City, CA, USA). The thermal reaction settings used were 50 °C for 2 min, 95 °C for 2 min, and then 40 cycles of 95 °C for 15 s and 60 °C for 2 min. The forward (FW) and reverse (RV) primers used in this study were obtained from Biomers.net (Ulm, Germany) and can be found in Appendix A. A cycle threshold (Ct) value was generated by setting the threshold during the geometric phase of cDNA sample amplification. The relative expression of each gene was calculated according to cyclophilin peptidylprolyl isomerase A (*Ppia*) mRNA levels and normalized to the levels measured in the control group. The ΔΔ*C*t method was used and corrected for primer efficiency [35]. Only samples with a quantification cycle lower than 30 were used for fold-change calculation. Melt curves for all qPCR products were checked to ensure a single PCR product was generated and are included in Appendix A.

### 4.8. Western Blotting Analysis

HepG2 cells were harvested and homogenized in 300 µL RIPA lysis buffer (50 mM Tris-HCl pH 7.4, 150 mM NaCl, 1% NP-40, and 0.25% Na-deoxycholate containing protease and phosphatase inhibitors). The total protein content was measured using the Pierce BCA protein assay kit (Thermo Scientific, Madrid, Spain). Samples were denatured by mixing with a loading buffer solution (0.5 M Tris-HCl pH 6.8, glycerol, SDS, β-mercaptoethanol, and bromophenol blue) and then heated at 99 °C for 5 min using a thermocycler (Multigene Labnet, Barcelona, Spain). Acrylamide gels were prepared using a TGX Fast Cast Acrylamide Kit, 10% (Bio-Rad, Barcelona, Spain), and 25 µg of protein was subjected to SDS-polyacrylamide gel electrophoresis (PAGE) using electrophoresis buffer (192 mM glycine, 25 mM Tris base, and 1% SDS). Proteins were electrotransferred onto supported PVDF membranes (Trans-Blot Turbo Mini PVDF Transfer Packs, Bio-Rad, Barcelona, Spain). After blocking with 5% non-fat dried milk, the membranes were incubated with gentle agitation overnight at 4 °C with specific antibodies against pSTAT3 (ab68153; Abcam, Cambridge, UK) and ObRb (ab177469; Abcam) diluted 1:1000. To analyse the expression of β-actin as a loading control, the membranes were incubated with a rabbit anti-actin primary antibody (A2066; Sigma Aldrich, Madrid, Spain) diluted 1:1000. Finally, membranes were incubated with anti-rabbit horseradish peroxidase secondary antibody (NA9344, GE Healthcare, Barcelona, Spain) diluted 1:10,000. Protein levels were detected with the chemiluminescent detection reagent ECL Select (GE Healthcare, Barcelona, Spain) using GeneSys image acquisition software (G:Box series, Syngene, Barcelona, Spain). Finally, protein bands were quantitated by densitometry using ImageJ software (W. S. Rasband, Bethesda, MD, USA) and each band was normalized to the corresponding β-actin band. Finally, the treatment groups were normalized to the control group.

### 4.9. Immunofluorescence Analysis

The ObRb content in steatotic HepG2 cells treated with 10 and 50 µM resveratrol for 24 h was visualized by immunofluorescent detection. After fixation, the cells were blocked with 1% BSA in PBST for 1 h at room temperature and incubated with primary anti-leptin receptor antibody (5 µg/mL, ab60042; Abcam) in a humidified chamber overnight at 4 °C. After washing with PBS, the cells were incubated with the secondary antibody (1/250, ab96899; Abcam) for 1 h at room temperature in the dark. Subsequently, the cells were exposed to 1 µg/mL 4,6-diamidino-2-phenylindole (DAPI, Thermo Scientific, Madrid, Spain) for 1 min at room temperature, mounted with Vectashield (Vector Laboratories, Burlingame, CA, USA) and sealed with nail polish to prevent drying and movement under a microscope. Images were taken using a Nikon Eclipse TE2000-E laser scanning confocal microscope at 60× magnification.

### 4.10. UHPLC-MSn Analysis of Cell Media

Cell media supernatants were collected and analysed by UHPLC-MSn to determine the stability and cellular metabolism of RSV in cell culture. Cell media was extracted according to Sala et al. [36] and analysed according to previous reports [7,37]. Briefly, a mixture of 150 µL cell media and 150 µL cold methanol acidified with formic acid 1% (*v*/*v*) was vortexed for 1 min. The mixture was then centrifuged at 12,000 rpm for 5 min at room temperature. The supernatant was directly injected into the UHPLC-MSn system. Samples were analysed by an Accela UHPLC 1250 coupled to a linear ion trap–mass spectrometer (LTQ XL, Thermo Fisher Scientific Inc; San Jose, CA, USA) fitted with a heated electrospray ionization source (H-ESI-II; Thermo Fisher, Madrid, Spain). Data processing was performed using Xcalibur software from Thermo Scientific. Metabolite identification was performed by comparing the retention time with the MS fragmentation patterns with authentic standards in negative ionization mode. Quantification was carried out using specific MS^2^ full scans and calibration curves for pure standards [37].

### 4.11. Statistical Analysis

Data have been expressed as the mean ± SD of at least three independent assays for each experiment. The data were evaluated by Student’s *t*-test to identify significant differences between the controls and the treatments. Outliers were determined by Grubbs’ test. The statistical analyses were performed using XLSTAT 2017 (Addinsoft, Paris, France, 2017). Graphics were made using GraphPad Prism 6 (GraphPad Software, San Diego, CA, USA). Differences were considered significant when *p* values were less than 0.05 and differences were considered as indicating a tendency when *p* values were less than 0.1.

## 5. Conclusions

In the current study, we established the experimental conditions under which to develop a cellular model of leptin-resistance in HepG2 cells and we demonstrate, for the first time, the ability of RSV to restore the altered HepG2 cellular response to leptin by enhancing the protein content of the leptin receptor OBRb without any significant change in SIRT1 activity. Hence, leptin action is crucial for the maintenance of a correct lipid metabolism in hepatic cells, and we suggest the use of RSV as a natural agent in the formulation of functional foods to restore leptin signalling and subsequently combat the onset of obesity.

## Figures and Tables

**Figure 1 ijms-20-06282-f001:**
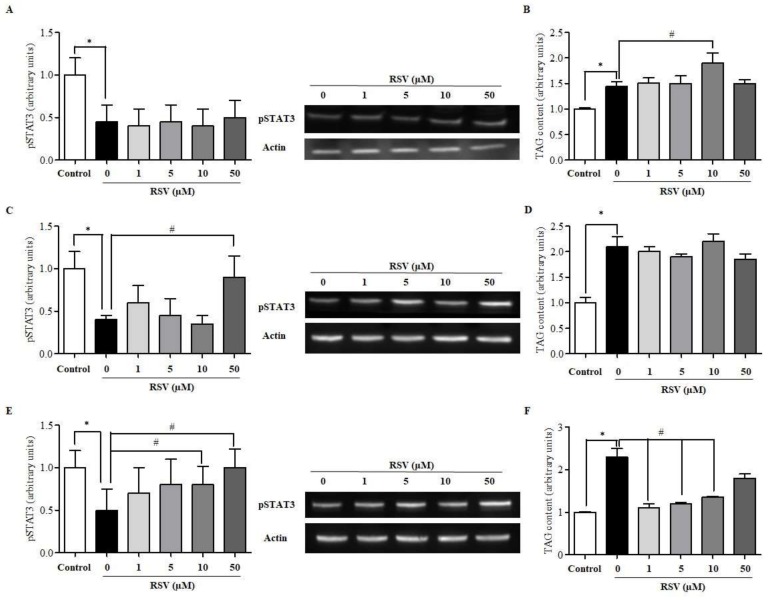
Impact of resveratrol (RSV) on leptin signalling in human hepatoma (HepG2) cells. pSTAT3 activation and triglyceride (TAG) content were determined in palmitate (Palm)-induced steatotic HepG2 cells treated with RSV for (**A**) and (**B**) 20 min, (**C**) and (**D**) 6 h, and (**E**) and (**F**) 24 h. Leptin signalling was activated by the addition of 10 ng/mL leptin during the last 20 min of incubation. * indicates p < 0.05 when comparing the 0 µM RSV group to the control group and # indicates p < 0.05 when comparing 1, 5, 10 or 50 µM RSV groups to the 0 µM RSV group as assessed by Student’s t-test. Data are expressed as the mean ± SD of three replicates. A representative WB image for each RSV incubation time is also included.

**Figure 2 ijms-20-06282-f002:**
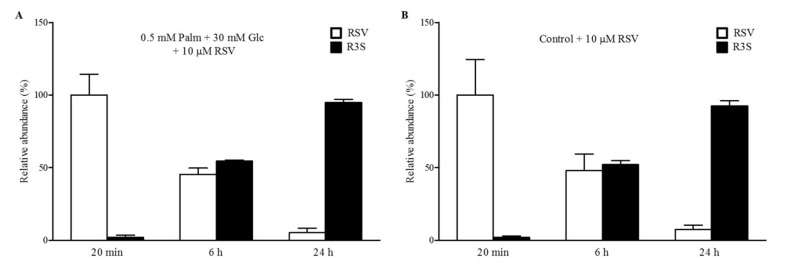
Metabolization of RSV in HepG2 cells. The graphic shows the conversion of 10 μM RSV to RSV-3-sulfate (R3S) in (**A**) steatotic cells and (**B**) control cells treated for different time periods. Cell media supernatants were collected and analysed by ultra-high-performance liquid chromatography–mass spectrometry (UHPLC-MSn) to determine the stability and cellular metabolism of RSV in cell culture. Data are expressed as the mean ± SD of three replicates.

**Figure 3 ijms-20-06282-f003:**
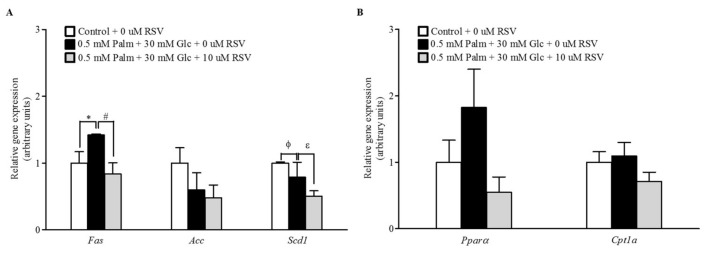
Effects of RSV on lipid metabolism in steatotic HepG2 cells. The mRNA expression levels of (**A**) lipogenic and (**B**) lipolytic genes were assessed by qPCR after 24 h of supplementation with 10 μM RSV. Leptin signalling was activated by the addition of 10 ng/mL leptin during the last 20 min of incubation. * *p* < 0.05 and ^ϕ^
*p* < 0.1 comparing the Palm + Glc group to the control group and ^#^
*p* < 0.05 and ^ε^
*p* < 0.1 comparing the RSV group to the Palm + Glc group as assessed by Student’s *t*-test. Data are expressed as the mean ± SD of three replicates. Legend: *Fas*, fatty acid synthase; *Acc*, acetyl-CoA carboxylase; *Scd1*, stearoyl-CoA desaturase-1; *Pparα*, peroxisome proliferator-activated receptor α; *Cpt1a*, carnitine palmitoyltransferase 1a.

**Figure 4 ijms-20-06282-f004:**
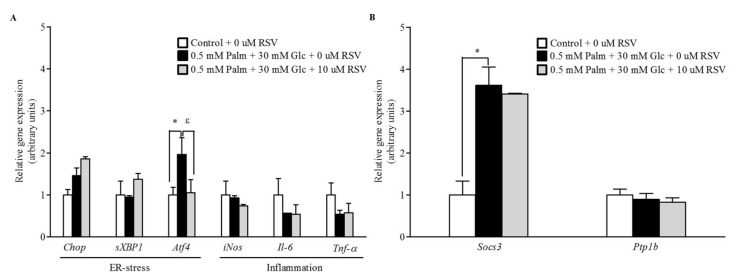
Mechanisms of action of RSV in enhancing the response to leptin (I). Steatotic HepG2 cells were treated with 10 μM RSV for 24 h. Leptin signalling was activated by the addition of 10 ng/mL leptin during the last 20 min of incubation. mRNA gene expression of (**A**) endoplasmic reticulum (ER) stress and inflammation markers and (**B**) inhibitors of the leptin signalling pathway were assessed by qPCR. * *p* < 0.05 comparing the Palm + Glc group to the control group and ^ε^
*p* < 0.1 comparing the RSV group to the Palm + Glc group as assessed by Student’s *t*-test. Data are expressed as the mean ± SD of three replicates. Legend: *Chop*, DNA damage inducible transcript 3; *sXbp1*, spliced x-box binding protein 1; *Atf4*, activating transcription factor 4; *Il-6*, interleukin-6; *iNos*, inducible nitric oxide synthase, *Socs3*, suppressor of cytokine signalling 3; *Ptp1b*, tyrosine-protein phosphatase non-receptor type 1; *Tnf-α*, tumour necrosis factor alpha.

**Figure 5 ijms-20-06282-f005:**
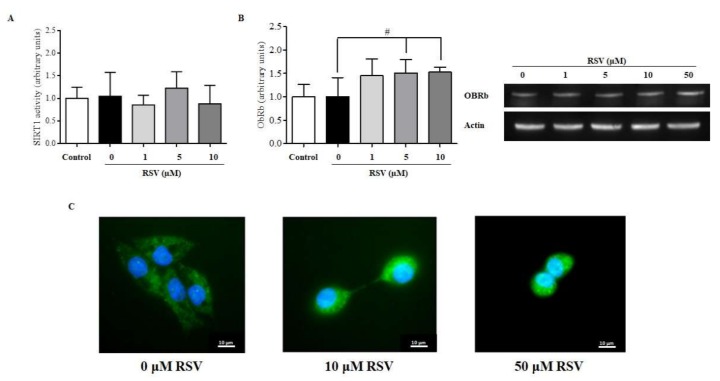
Mechanisms of action of RSV in enhancing the response to leptin (II). (**A**) Sirtuin-1 (SIRT1) enzyme activity and (**B**) hepatic longest receptor isoform of leptin (OBRb) protein content as assessed using WB. A representative WB image for each RSV incubation time is included. (**C**) shows a representative immunofluorescent image showing the OBRb content (green) of HepG2 cells treated with 0, 10, and 50 μM RSV for 24 h. ^#^
*p* < 0.05 when comparing the RSV groups to the Palm + Glc group as assessed by Student’s *t*-test. Data are expressed as the mean ± SD of three replicates.

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
