# Peer review of "Resveratrol Treatment Enhances the Cellular Response to Leptin by Increasing OBRb Content in Palmitate-Induced Steatotic HepG2 Cells"

_ijms, 2019, doi:10.3390/ijms20246282_

Round 1

Reviewer 1 Report

This manuscript is very well written, but before this manuscript will be accepted for publication the authors should improved this manuscript:

In introduction is not enough information concerning properties of resveratrol. Please write some more information about this compound, for example: structure of it, important biological properties of resveratrol. I don't see the conclusions of yours work. Please write the conclusions after Statistical analysis. The end of discusion in your manuscript is better for chapter conclusions than discusion.

Author Response

Reviewer #1:

This manuscript is very well written, but before this manuscript will be accepted for publication the authors should improved this manuscript:

In introduction is not enough information concerning properties of resveratrol. Please write some more information about this compound, for example: structure of it, important biological properties of resveratrol. I don't see the conclusions of yours work. Please write the conclusions after Statistical analysis. The end of discusion in your manuscript is better for chapter conclusions than discusion.

According to the Reviewer comment, new data on resveratrol bioactivity and chemical structure has been incorporated in the Introduction section of the revised Manuscript (page 1-2, lines 43-49). Specifically, the following paragraph has been added accordingly:

In addition, resveratrol (3,5,4’-trihydroxystilbene, RSV), a dietary non-flavonoid polyphenol found in grapes (0.002-0.008 mg/g) and red wine (1.98-7.13 mg/L), also protected the liver against lipid metabolic disorders in various rodent models of hepatic steatosis, possibly via regulating SIRT1 activity and endoplasmic reticulum (ER) stress [5–7]. In fact, RSV has been indicated to obtain a wide range of biological effects including cardioprotective, anticancer, anti-inflammatory and antioxidant properties [8,9]. However, the exact mechanisms by which RSV exerts its beneficial effect are still unclear and require more research.

The following references have been also included in the Introduction section of the revised Manuscript:

[8] Baur, J.A; Sinclair, D.A. Therapeutic potential of resveratrol: the in vivo evidence. Nat Rev Drug Discov. 2006, 5, 493–506.

[9] Baur, J.A; Pearson, K.J; Price, N.L; Jamieson, H.A; Lerin, C; Kalra, A; Prabhu, V.V; Allard, J.S; Lopez-Lluch, G; Lewis, K; Pistell, P.J; Poosala, S; Becker, K.G; Boss, O; Gwinn, D; Wang, M; Ramaswamy, S; Fishbein, K.W; Spencer, R.G; Lakatta, E.G; Le Couteur, D; Shaw, R.J; Navas, P; Puigserver, P; Ingram, D.K; de Cabo, R; Sinclair, D.A. Resveratrol improves health and survival of mice on a high-calorie diet. Nature. 2006, 444, 337–342.

In addition, as indicated by the Reviewer, the end of discussion has been modified accordingly in the revised Manuscript and a new section have been incorporated after the Statistical Analysis paragraph (page 10, lines 384-390):

“5. Conclusions: In the current study, we established the experimental conditions to develop a cellular model of leptin-resistance in HepG2 cells and we demonstrate, for the first time, the ability of RSV to restore the altered HepG2 cellular response to leptin by enhancing the protein content of the leptin receptor OBRb without any significant change in SIRT1 activity. Therefore, leptin action is crucial for the maintenance of correct lipid metabolism in hepatic cells, and we suggest the use of RSV as a natural agent in the formulation of functional foods to restore leptin signalling and subsequently combat the onset of obesity”.

Reviewer 2 Report

The authors have investigated in a steatosis in vitro model the restoration of leptin intracellular signalling under the influence of a natural compound, resveratrol in a steatosis. The study is complex and offers new and valuable data concerning possible mechanisms of reducing leptin resistance, with wide implications not only in non-alcoholic steatosis but also in metabolic syndrome and obesity. However, a few issues need to be discussed more thoroughly in the manuscript:

The authors should explain in the Discussion section of the manuscript how their data obtained from a cancerous cell line (HepG2) may be translated to normal human cells. For example, in the human body, OBR b receptors are normally expressed in the hypothalamus while the "short" OBR a,c,d,f are expressed peripherically in a variety of tissues. It is true that HepG2 cells express OBRb receptors, but are there enough studies to confirm that this "long" receptor is significantly present on normal hepatocytes? In Lines 198-199 of the manuscript the authors state  "the capacity of RSV to downregulate Fas and SCD1 gene expression". A recently published study (Hackel et al, Nature Commun, 2019, 10, 2717) proved in vivo in a rodent model that it is leptin who supresses hepatic de novo synthesis by downregulating the expression of Fas and SCD1. Your study actually confirmed this finding, probably it is an indirect effect of resveratrol who augments the expression of leptin and restores its signalling. Your data concerning the lack of modifications concerning the expression of genes encoding PPAR alpha confirmed the study mentioned above which similarly found that the reduction of lipogenesis caused by leptin was not due to hepatic beta-oxidation. You should definitely add this new study in your Discussion section and correlate it to your findings.

Author Response

The authors have investigated in a steatosis in vitro model the restoration of leptin intracellular signalling under the influence of a natural compound, resveratrol in a steatosis. The study is complex and offers new and valuable data concerning possible mechanisms of reducing leptin resistance, with wide implications not only in non-alcoholic steatosis but also in metabolic syndrome and obesity. However, a few issues need to be discussed more thoroughly in the manuscript:

The authors should explain in the Discussion section of the manuscript how their data obtained from a cancerous cell line (HepG2) may be translated to normal human cells. For example, in the human body, OBR b receptors are normally expressed in the hypothalamus while the "short" OBR a,c,d,f are expressed peripherally in a variety of tissues. It is true that HepG2 cells express OBRb receptors, but are there enough studies to confirm that this "long" receptor is significantly present on normal hepatocytes?

We completely agree with the Reviewer that this is an important point to be carefully discussed and valued. Therefore, the following limitation statement has been incorporated in the Discussion section of the revised Manuscript (page 7, line 240-244):

In this context it is also important to point out that although there could be considerable differences in OBRb content between cancerous and non-cancerous cells, with regard to regulation of hepatic metabolism, HepG2 cells appear to be closer to the in vivo situation despite the tumorigenic origin [27]. However, since we documented this only for HepG2 cells, future studies are needed to evaluate whether this mechanism is also at work in non-cancerous hepatic cells.

In addition, the following reference has been also included:

[27] Sefried, S; Häring, H.U; Weigert, C; Eckstein, S.S. Suitability of hepatocyte cell lines HepG2, AML12 and THLE-2 for investigation of insulin signalling and hepatokine gene expression. Open Biol. 2018, 8, 180147

In Lines 198-199 of the manuscript the authors state "the capacity of RSV to downregulate Fas and SCD1 gene expression". A recently published study (Hackel et al, Nature Commun, 2019, 10, 2717) proved in vivo in a rodent model that it is leptin who supresses hepatic de novo synthesis by downregulating the expression of Fas and SCD1. Your study actually confirmed this finding, probably it is an indirect effect of resveratrol who augments the expression of leptin and restores its signalling. Your data concerning the lack of modifications concerning the expression of genes encoding PPAR alpha confirmed the study mentioned above which similarly found that the reduction of lipogenesis caused by leptin was not due to hepatic beta-oxidation. You should definitely add this new study in your Discussion section and correlate it to your findings.

We really appreciate this comment and, therefore, we have incorporated this new study in the Discussion of the revised Manuscript (page 6, lines 199-204). Specifically, the following paragraph has been added accordingly:

In this sense, a recent study demonstrated the ability of brain leptin to protect from hepatic steatosis by decreasing de novo lipogenesis in the rat liver [21]. In fact, our results indirectly confirmed this finding, probably due to the effect of RSV on increasing the cellular response to leptin. In addition, the lack of leptin-induced changes in hepatic gene expression of Pparα and Cpt1a, which are regarded as important regulators of lipid oxidation, did not suggest a major role for hepatic β-oxidation in our experimental treatment with RSV

[21] Hackl MT, Fürnsinn C, Schuh CM, Krssak M, Carli F, Guerra S, Freudenthaler A, Baumgartner-Parzer S, Helbich TH, Luger A, Zeyda M, Gastaldelli A, Buettner C, Scherer T. Brain leptin reduces liver lipids by increasing hepatic triglyceride secretion and lowering lipogenesis. Nat Commun. 2019, 10, 2717.